# Combined Supplementation of Nano-Zinc Oxide and Thyme Oil Improves the Nutrient Digestibility and Reproductive Fertility in the Male Californian Rabbits

**DOI:** 10.3390/ani10122234

**Published:** 2020-11-27

**Authors:** Ahmed A. A. Abdel-Wareth, Mohammed Ali Al-Kahtani, Khalid Mushabab Alsyaad, Fatma Mohsen Shalaby, Islam M. Saadeldin, Fahdah Ayed Alshammari, Muhammad Mobashar, Mohamed H. A. Suleiman, Abdalla H. H. Ali, Mohamed O. Taqi, Hazem G. M. El-Sayed, Mahmoud S. Abd El-Sadek, Abdallah E. Metwally, Ahmed Ezzat Ahmed

**Affiliations:** 1Department of Animal and Poultry Production, Faculty of Agriculture, South Valley University, Qena 83523, Egypt; A.wareth@agr.svu.edu.eg (A.A.A.A.-W.); attai73@yahoo.com (A.H.H.A.); 2Department of Biology, College of Science, King Khalid University, Abha 61413, Saudi Arabia; dr.malkahtani@gmail.com (M.A.A.-K.); alsyaad55@hotmail.com (K.M.A.); fshalabi@kku.edu.sa (F.M.S.); 3Director of Research Centre, College of Science, King Khalid University, Abha 61413, Saudi Arabia; 4Department of Zoology, Faculty of Sciences, Mansoura University, 35516 Mansoura, Egypt; 5Department of Physiology, Faculty of Veterinary Medicine, Zagazig University, Zagazig 44519 , Egypt; isaadeldin@ksu.edu.sa; 6Department of Animal Production, College of Food and Agricultural Science, King Saud University, Riyadh 11451, Saudi Arabia; 7Department of Biology, College of Sciences and Literature, Northern Border University, Rafha 76312, Saudi Arabia; fahdah.ayed@nbu.edu.sa; 8Department of Animal Nutrition, University of Agriculture, Peshawar 25000, Pakistan; mobashar@aup.edu.pk; 9Department of Chemistry, College of Science, King Khalid University, Abha 9004, Saudi Arabia; mhadam@kku.edu.sa; 10Central Laboratory for Agricultural Climate, Agricultural Research Centre, Ministry of Agriculture and Land Reclamation, Dokki-Giza 12311, Egypt; mohamed.o.taqi@gmail.com; 11Regional Centre for Food and Feed (RCFF), Agricultural Research Centre, Cairo 12619, Egypt; elgafary1987@gmail.com; 12Department of Physics, Faculty of Science, South Valley University, Qena 83523, Egypt; mahmoud.abdelsadek@sci.svu.edu.eg; 13Department of Nutrition and Clinical Nutrition, Faculty of Veterinary Medicine, Zagazig University, Zagazig 44511, Egypt; drabdalla75@yahoo.com; 14Department of Theriogenology, Faculty of Veterinary Medicine, South Valley University, Qena 83523, Egypt

**Keywords:** nano-zinc, nutrient digestibility, thyme oil, semen characteristics, testosterone

## Abstract

**Simple Summary:**

The present study aimed to investigate the beneficial effects of supplementing zinc oxide nanoparticles (ZnO-NPs), thyme oil (THO), or their combination on the reproductive parameters, nutrient digestibility coefficients, and some blood metabolites of male Californian rabbits. Results showed an improvement in the nutrients digestibility, liver and kidney functions, testosterone concentration, and semen characteristics (semen volume, sperm motility, vitality, and morphology). ZnO-NPs were more effective in improving the male fertility, while THO showed a marked improvement in nutrient digestibility. Therefore, the inclusion of ZnO-NPs, or THO, or both is recommended for a rabbit feeding regimen to improve the feeding profitability.

**Abstract:**

The present study aimed to determine the effects of zinc oxide nanoparticles (ZnO-NPs), thyme oil (THO), or their combination on the nutrient digestibility coefficients, reproductive parameters, and some blood metabolites of male Californian rabbits. One hundred rabbits, 29-weeks of age (initial body weight 3.48 ± 0.08 kg) were randomly distributed into four groups, 25 rabbits each. Treatment groups were fed a control diet, a control diet supplemented with ZnO-NPs (100 mg/kg), THO (500 mg/kg), or combination of ZnO-NPs (100 mg/kg) and THO (500 mg/kg). The feeding trial lasted for 35 days. Results showed improvements in dry matter, crude protein, ether extract, and crude fiber in ZnO-NPs, THO, and their combination treated groups compared to those of control. Furthermore, semen volume, sperm motility, vitality, and morphology were significantly improved (*p* < 0.01) in ZnO-NPs and THO groups rather than the control. Both ZnO-NPs and THO, as either individual or combined treatments significantly improved the serum alanine amino-transferase (ALT), aspartate amino-transferase (AST), urea, and creatinine compared to the control. Moreover, serum concentrations of testosterone were significantly increased in rabbits supplemented with ZnO-NPs, THO, or their combination compared to those of control (*p* < 0.05). In conclusion, ZnO-NPs, THO, or their combination improved the digestibility of nutrients, liver/ kidney functions, semen characteristics, and testosterone concentration in male rabbits.

## 1. Introduction

Feed additives of trace minerals in nanoparticles were recently used to improve the productive and reproductive parameters of farm animals [1,2]. The nano-scaled size particles, e.g., nano zinc-oxide (ZnO-NPs; 1–100 nm), provided novel physical, chemical, and biological properties [3,4]. Zinc plays a pivotal role in the sperm cell functions, including lipid flexibility, cell membrane stabilization [5], sperm capacitation, and acrosomal reaction [6]. Moreover, it was found to increase the semen volume, total live sperm count, sperm motility and conception in heat-stressed rabbits [7], while its deficiency exerted oxidative damages, infertility, and worse sperm quality due to impaired spermatogenesis, lowered testosterone secretion, and increased sperm abnormalities [8]. Zinc supplementation was used to improve the sperm count, motility, form, function, quality, and fertilizing capacity [1]. The commonly used grains in rabbit diets are rich in phytate that may reduce or inhibit absorption of zinc [9,10].

Thyme (*Thymus vulgaris* L.) improves the growth rate, antioxidant activity, and appetite promotion [11,12,13,14,15,16]. Likewise, supplementation with thyme oil (THO; 0.5 g/kg) was found to improve the intestinal integrity and total antioxidant status of rabbits [17]. The main components of thyme oil are thymol, carvacrol, p-cymene, g-terpinene, linalool, b-myrcene, and terpinen-4-ol [13,18]. Those constituents were known to have antioxidant properties [19] and may improve liver and kidney functions and abdominal fat accumulation [13,20].

The present study investigates the effects of zinc in nanoparticle form and thyme essential oils for improving the reproductive parameters, nutrient digestibility coefficients, and some blood metabolites of the male Californian rabbits.

## 2. Materials and Methods

### 2.1. Experimental Animals, Design and Management

The study was conducted at the rabbit research farm, Department of Animal and Poultry Production, Faculty of Agriculture, South Valley University, Qena, Egypt. A completely randomized design was used for investigating effects of ZnO-NPs, THO and their combination. A total of 100 of 29-weeks-old male Californian rabbits (initial body weight 3.48 ± 0.08 kg) were randomly distributed into four equal groups (*n* = 25). Treatment groups were fed a control diet, a control diet supplemented with ZnO-NPs (100 mg/kg), THO (500 mg/kg), or a combination of ZnO-NPs (100 mg/kg) and THO (500 mg/kg). The treatments were continuing for 35 days. Animals were individually reared in a closed building in cages (measuring 48 cm × 55 cm × 38 cm, for width × length × height, respectively) of galvanized wire net, equipped with an automatic drinkers and manual feeders. Ambient temperature was maintained at 22 °C with a 12-h light/dark cycle. The feed and fresh tap water was available *ad libitum*. The basal diet and its feeding value were presented in Table 1. Rabbits were managed for the experiment according to the animal rights and ethical committee in South Valley University, Egypt. The experiment protocol (SVUAGR.42018) was approved by the Animal Ethics Committee of Animal and Poultry Department, South Valley University in accordance with the guidelines of Egyptian Research Ethics Committee and the guidelines contained in the Guide for the Care and Use of Laboratory Animals (2011).

### 2.2. Zinc Oxide Nanoparticles and Thyme Oil Preparation

Zinc oxide nanoparticles (Cas no.1314-13-2, Sigma-Aldrich, Steinheim, Germany) concentration was 20 wt. % in distillated water, and their size is less than 40 nm. Using dynamic light scattering technique, the particle hydrodynamic diameter was less than 100 nm. The pH was 7.5 ± 1.5 (for aqueous systems) and density was 1.7 ± 0.1 g/mL 60.1 g/mL at 25 °C [21]. The structural morphology of the particle was performed through scanning and transmission electron microscope (SEM, TEM) (Figure 1). The chemical compounds of the hydrodistilled THO (Table 2) was analyzed through a gas chromatography-spectrometry (GC-MS; Sigma 300 apparatus attached to a HP of 5970 300 mass spectrometer) system. Thyme essential oils (TEO) were analyzed by gas chromatography (Delsi 121C gas chromatograph) according to Abozid and Asker [22] (Table 2, Figure 2).

### 2.3. Nutrients Digestibility Trial

At the end of the experiment at 33 weeks of age, a digestibility trial was carried out for four days. During the 4-day period trial, daily feed intake was determined and total faeces from each replicate were collected, weighed, and frozen at −10 °C until preparation for chemical analysis. Faecal samples were partially dried at 60 °C for 48 h before being analyzed. Samples of the feed and faeces were analyzed for moisture by oven drying (930.15), ash by incineration (942.05), protein by Kjeldahl (984.13), and ether extract by Soxhlet fat analysis (954.02), calcium (927.02) and phosphorous (935.59) as described by the AOAC International (2006). Gross energy was determined using adiabatic bomb calorimetry (Parr Instrument Company, IL, USA). The digestibility of nutrients was calculated according to the following equation: Digestibility, % = [(t − f)/t] × 100, where *t* is the nutrient intake during the collection period [g] and ƒ is the amount of nutrient excreted in faeces [g]. At the end of the experiment at 33 weeks of age, a digestibility trial was carried out for four days according to Perez et al. [23]. The DE refers to GE intake minus energy lost in faeces according to Hall et al. [24].

### 2.4. Semen Collection and Evaluation

Quality of semen ejaculates of the male rabbits was assessed as described by El-Desoky et al. [25]. The ejaculates were collected by using artificial vagina, in graduated collection tubes after removal of the gel mass. Semen was collected from 25 animals (one ejaculate from each animal and per treatment at 34 weeks of age. Assessments of live, dead, and abnormal sperms were performed by counting 200 sperm cells using an Eosin-Nigrosin staining mixture. Complete or partial purple- stained sperm cells were considered non-viable, whereas non-stained sperm cells were considered viable. Percentages of motile sperms at the warm stage showing progressive forward movement were visually calculated in several microscopic fields under 100× magnifications using light microscope.

### 2.5. Blood Biochemical Assay

At the end of the experimental period, the treated animals were anaesthetized by intramuscular injection of ketamine and xylazine, and then 5 mL blood was withdrawn from one of the marginal ear veins. Blood samples were centrifuged at 1008× *g* for 15 min, where the serum was collected and stored at −20 °C until assayed for biochemical analysis. Serum testosterone concentrations were measured by immunoassay using commercial kits (Monobind Inc., Lake Forest, CA, USA). Liver enzymes; alanine transaminase (ALT) and aspartate transaminase (AST), as well as the kidney function markers; Creatinine and urea, concentrations were also measured by using standard diagnostic kits (Monobind Inc. Lake Forest, CA, USA).

### 2.6. Statistical Analysis

All data were analyzed using one-way analysis of variance (ANOVA) by using GraphPad Prism Software Version 3 (GraphPad Prism, San Diego, CA, USA). The significant differences between the treatments were confirmed by Tukey as a post-hoc test and distinguished by Duncan’s multiple range [26] as letters on bars (a, b, c). All values were expressed as mean ± SEM for 25 animals. The differences between groups were considered significant at *p* < 0.05.

## 3. Results

### 3.1. Nutrient Digestibility Coefficients

Effects of dietary nano-zinc oxide (ZnO-NPs), thyme oil (THO), or their combination; ZnO-NPs +THO, on nutrients digestibility coefficients were shown in Table 3. All treatments, ZnO-NPs, THO, and ZnO-NPs +THO, significantly increased the nutrients’ digestibility compared to each respective control; dry matter (68.38 ± 0.71 vs. 64.70 ± 1.02%), crude protein (85.56 ± 0.51 vs. 78.06 ± 1.13%), Ether extract (88.40 ± 1.46 vs. 80.98 ± 0.89%), and crude fiber (27.91 ± 0.61 vs. 24.92 ± 1.67%) (*p* < 0.05).

### 3.2. Semen Quality

Supplementation of ZnO-NPs, THO or ZnO-NPs + THO effects on fertility in terms of semen volume, sperm motility, vitality, and morphology of the male rabbits were shown in Table 4. All treatments significantly improved the semen quality compared to that of control, like; increased sperm vitality, motility, and semen volume in variable potencies (*p* < 0.001). Abnormal sperm cells in response to all treatments were significantly lowered compared to those of control (*p* < 0.001). The ZnO-NPs were the best in decreasing the sperm shape abnormality (15.58 ± 0.51%), while THO could be the best stimulating factor of sperm motility (75.67 ± 0.86%). Although, no significant differences were detected between the treated groups, the combined treatment of ZnO-NPs and THO tended to be the most effective factor in increasing the sperm vitality (84.50 ± 0.70%). The ZnO-NPs or THO significantly increased the volume of semenal plasma compared to the controls (*p* < 0.05), while the combined ZnO-NPs + THO highly increased the volume of semen compared to controls (*p* < 0.0001) rather than either ZnO-NPs or/ THO each alone (*p* < 0.05).

### 3.3. Serobiochemical Assays

Effects of ZnO-NPs, THO, or their combination on the liver and kidney function markers as well as testosterone levels in serum of the tested rabbits are shown in Figure 3, Figure 4 and Figure 5, respectively. Rabbits received either ZnO-NPs, THO, or their combination; ZnO-NPs + THO showed the lower serum levels of ALT (17.4 ± 0.06, 17.0 ± 0.51, 16.8 ± 0.50 IU/L, respectively) compared to control (23.2 ± 0.50 IU/L) (*p* < 0.0001) (Figure 3A). The serum concentrations of AST in response to ZnO-NPs, THO, or ZnO-NPs + THO were also significantly decreased (17.8 ± 0.40, 17.1 ± 0.5, 16.9 ± 0.80 IU/L, respectively) compared to control (26.3 ± 0.3 IU/L) (*p* < 0.0001) (Figure 3B).

Likewise, serum creatinine and urea were significantly decreased in response to ZnO-NPs, THO, or; ZnO-NPs + THO compared to the controls (*p* < 0.0001) (Figure 4). The serum concentrations of creatinine (Cr) in response to ZnO-NPs, THO, or ZnO-NPs + THO were significantly decreased (0.93 ± 0.20, 0.98 ± 0.02, 1.18 ± 0.05 mg/dl, respectively) compared to control (1.55 ± 0.03 mg/dl) (*p* < 0.0001) (Figure 4A). Furthermore, the serum concentrations of urea in response to ZnO-NPs, THO, or ZnO-NPs + THO were significantly decreased (36.7 ± 0.60, 37.2 ± 0.40, 38.0 ± 0.20 mg/dl, respectively) rather than those of controls (44.1 ± 0.10 mg/dl) (*p* < 0.0001) (Figure 4B).

All treatments, ZnO-NPs, THO, or the combined ZnO-NPs + THO, significantly increased the serum concentrations of testosterone (T) (4.2 ± 0.60, 2.3 ± 0.05, 8.20 ± 0.70 ng/mL, respectively) compared to control (0.45 ± 0.01 ng/mL) in variant potencies (*p* < 0.05–0.0001). However, the combined ZnO-NPs + THO treatment was the most effective rather than the CTL or the other treatment groups; ZnO-NPs or THO (Figure 5A). Sperm cells of the tested rabbits were stained with eosin-nigrosin showing unstained alive cells and those red-stained dead (Figure 5B).

## 4. Discussion

Dietary essential oils are digestive enhancers with specific antimicrobial activity which promotes the general health condition [14,27]. Thyme oil (THO) accordingly showed a beneficial effect on the rabbit performance, welfare, and health status especially in hot environments [13,17,28]. It was reported that thyme oil plays a pivotal role in stimulating the feed intake, increasing the body weight/gain and intestinal integrity, and thus improving the overall feed conversion ratio [13,17]. Those positive effects were attributed to the thyme compounds on the digestive efficiency [29]. Furthermore, zinc oxides in the form of nanostructure have become interesting areas of research in animal husbandry [30]. Due to scarcity of available reports on effect of zinc oxide nanoparticles on the male of rabbits, comparison was done with other studies that used other zinc sources. Nanoparticles exhibit novel properties, such as great specific surface area, high surface activity with high catalytic efficiency [31]. The mineral nanoparticles can increase absorption [32,33]. Zinc requirement for rabbits is 30–170 mg/kg dry matter, with higher requirements for breeders [34,35]. Zinc plays a pivotal role in carbohydrates, lipids and protein metabolism for energy utilization liberated from feeds digestion [36,37]. The mineral is approved as essential cofactor for the metabolic reactions and enzyme activity especially the carrier proteins regulating digestion [38]. It was found that higher proportion of Zn (170 mg/kg) had improved the nutrient digestibility of rabbits compared to non-treated control group [35]. Simultaneously, in this study, the nanoparticles of zinc oxide (ZnO-NPs; 100 mg/kg BW), thyme oil (THO; 500 mg/kg BW), or their combination significantly increased the dry matter, crude protein, ether extract, and crude fiber digestibility compared to each respective control group that could be attributed to improvement of enzymes‘ activities induced by zinc [39,40,41,42] in addition to the improved digestibility induced by thyme [43] as stated in other species.

Chandak et al. [44] found that THO has a synergistic effect with zinc oxide and showed higher levels of antimicrobial effect against the root canal pathogens. In present study, rabbits received either ZnO-NPs, THO, or their combination significantly lowered the serum levels of ALT, AST, urea, and creatinine when compared to the control group (*p <* 0.001). Furthermore, concentrations of testosterone in the serum increased significantly in those male rabbits received with ZnO-NPs, THO, or their combination when compared to the control (*p <* 0.05). Improvement in liver and kidney functions as well as that the male sex steroid, testosterone, is attributed to the role of Zn in many biochemical processes and physiological functions. It has been reported that Zn is required for the normal function of plentiful structural proteins, enzymes, and hormones which are necessary for growth and development [45]. The improved concentrations of testosterone in response to Zn-diets supplemented groups might be due to enhancing the Leydig cells of the testis for increasing the levels of testosterone production [7,46,47]. A combination of ZnO-NPs and THO improved liver and kidney function as well as serum testosterone of Californian male rabbits indicating a possible synergistic effect of ZnO-NPs and THO in improving testosterone secretion. The study showed that supplementation of the ZnO-NPs, THO, or their combination to rabbits exhibited a significant increase in the ejaculate volume, sperm viability, and sperm motility. No information is available regarding the effect of nano zinc oxide supplementation on semen quality and fertility of male rabbit. However, our results were consistent with Moce et al. [48] who stated increased numbers of spermatozoa of rabbit bucks fed on rations supplemented with zinc in amount of 35 to 100 mg/kg, as compared to their control group. Supplementation of Zn in doses of 50, 100, and 150 mg/kg to rabbit diet increased spermatozoa concentration of male rabbits [49]. Zn plays a pivotal role in the production of spermatozoa in parallel to the synthesis of DNA and RNA by enhancing the activity of DNA and RNA polymerases [50]. Furthermore, Zn supplementation was reported to duplicate the blood concentration levels of testosterone in adult men [51].

Acconcia and Marino [52] have reported that all the mammalian steroids including testosterone are biosynthesized from cholesterol. Testosterone is anabolic agent especially for the skeletal muscles and brain, and the trace amount of testosterone is synthesized in the adrenal cortex, but the profound production comes from the interstitial testicular Leydig cells [52]. Effects of ZnO-NPs and/or THO on blood testosterone could reflect their effects on cholesterol or other cellular and biochemical factors. Previous studies in other animal species stated that thyme extract [20] or ZnO-NPs [53] significantly lowered the blood concentrations of cholesterol. Accordingly, the blood cholesterol could be directed for the production of testosterone in agreement with our findings. Moreover, Zn concentrations in the Leydig cells were found to be significantly lowered in mice fed a Zn-deficient diet compared to controls, and that the zinc transporter-7 (ZnT7) was principally expressed in the Leydig cells ZnT7 which plays an important role in the regulation of testosterone synthesis by modulating the steroidogenic enzymes [54]. All above suggests that zinc could play a direct pivotal role on those testosterone-secreting cells which coincide with our findings for the highest increment of testosterone in response to ZnO-NPs supplementation. The effect of thyme on the blood testosterone concentrations was studied in male rats showing significant increase of the hormone in response to 5% thyme-supplemented diet compared to control throughout 28 days [55]. Therefore, supporting our results, coadministration of ZnO-NPs and THO synergistically increase the testosterone concentration rather than each alone. Two pathways could explain our findings; (a) testosterone synthesis from cholesterol induced by THO [20], or (a) endogenous molecular factors in the Leydig cells stimulated by the ZnO-NPs which require additional research and molecular studies [54]. The same pathway explains the increased secretion and volume of seminal plasma from the accessory genital glands which mainly depend on the testosterone secretion [56]. Those findings suggest that either ZnO-NPs or THO could stimulate the Leydig cells to secrete the synthesized testosterone, but the combined treatment, i.e., ZnO-NPs plus THO, may enhance both the synthesis and secretion of the hormone from its sources. Dietary supplementation with those additives compensate for the deficient nutrients required for male fertility and could resume the normal pathway of spermatogenesis, including the sperm cells morphology, vitality, and motility, in addition to stimulating the sex glands to produce much of the seminal plasma.

## 5. Conclusions

In view of the above findings, it can be concluded that ZnO-NPs, THO, or their combination in rabbit feeding resulted in improved nutrient digestibility, liver and kidney functions, as well as semen quality. The combined treatment of ZnO-NPs with THO suggested a possible synergistic effect for stimulating testosterone secretion and increasing the volume of seminal plasma. Further molecular analysis and fertility studies are required to elaborate on this combinational effect. 

## Figures and Tables

**Figure 1 animals-10-02234-f001:**
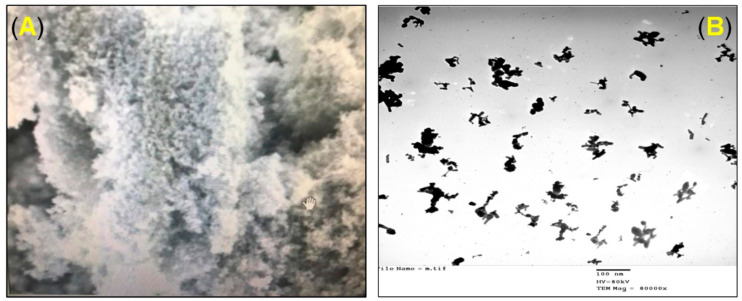
Scanning (SEM) (**A**), and transmission electron micrographs (TEM) (**B**) of synthesized zinc oxide nanoparticles (ZnO-NPs); 1–100 nm, HV = 80 kV, and TEM Mag = 8000×.

**Figure 2 animals-10-02234-f002:**
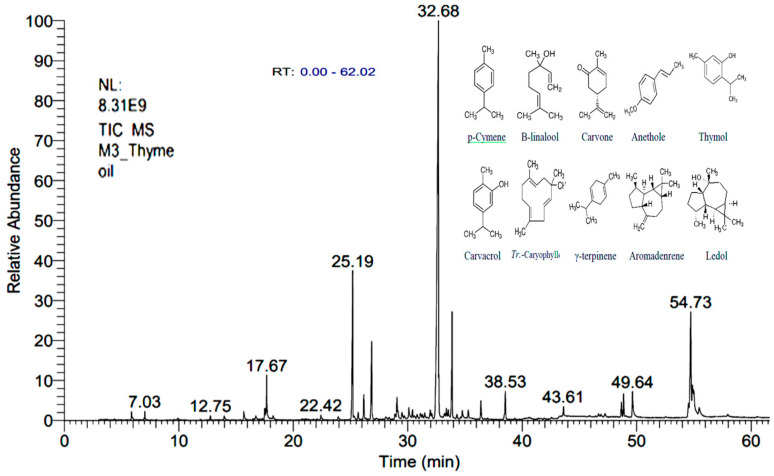
Retention time, area percent (%) and formulae of the major chemical compounds of essential oils of thyme as detected by the gas chromatography-mass spectrometry (GC-MS).

**Figure 3 animals-10-02234-f003:**
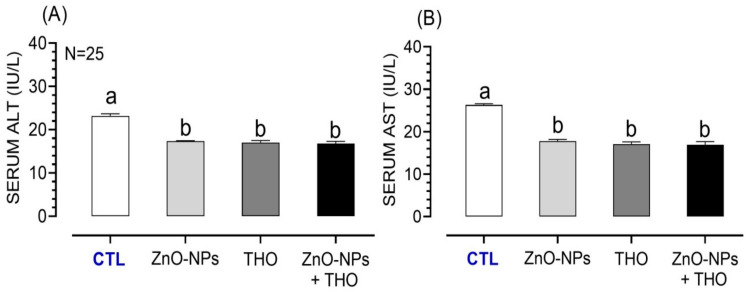
Serum concentrations of liver markers; alanine amino transferase (ALT) (**A**) and aspartate amino transferase (AST) (**B**) in response to supplementation of ZnO-NPs, THO, combined doses of both ZnO-NPs + THO, or a control (CTL) in male Californian rabbits. Letters on the bars a, b denote the significant difference between the different treatments (*p* < 0.05).

**Figure 4 animals-10-02234-f004:**
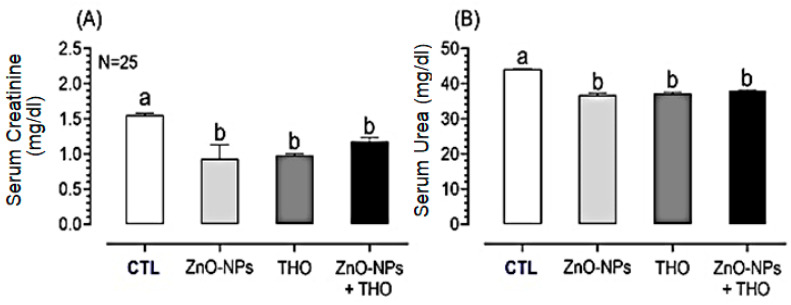
Serum concentrations of creatinine (Cr) (**A**) and urea (**B**) in response to supplementation of ZnO-NPs, THO, ZnO-NPs + THO, or a control (CTL) in male Californian rabbits, respectively. Other explanations were given in Figure 3.

**Figure 5 animals-10-02234-f005:**
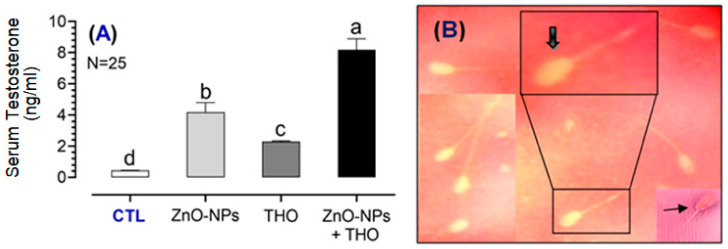
Serum concentrations of testosterone in response to supplementation with ZnO-NPs, THO, or ZnO-NPs + THO vs. control (CTL), respectively, is shown in (**A**). All treated groups significantly increased the testosterone concentrations with variable potencies, compared to control. Also, the combined treatment; ZnO-NPs was the most potent stimulator. Letters on bars (a, b, c, d) denote the significant differences between groups at *p* < 0.05. Alive, non-stained white (Thick arrow), and dead, red-stained sperm cell (thin arrow) are shown in (**B**). The red eosin stain could not penetrate the living sperm cells which appear white, but those dead spermatozoa are stained red with eosin, and so the sperm vitality could be distinguished and calculated.

**Table 1 animals-10-02234-t001:** Ingredient and chemical composition (as-fed basis) of the control diet fed to rabbits throughout the experimental periods.

Ingredients	%	Chemical Analysis	%
Yellow maize grain	32.00	Dry matter	91.40
Wheat bran	20.00	Ash	9.80
Soybean meal (44% CP)	18.00	Crude protein	17.00
Wheat straw	12.00	Crude fiber	12.60
Lucerne hay	5.00	Ether extract	2.90
Rice bran	5.00	Digestible energy (MJ/kg)	9.42
Linseed straw	2.80	Calcium	1.30
Sunflower meal	2.50	Phosphorus	0.86
Lime stone	2.00	Lysine	0.60
Sodium chloride	0.30	Methionine	0.41
Vitamin-mineral premix ^1^	0.30		
dl-Methionine	0.10		
Zinc	0.09		

^1^ Per kg of ration: vitamin A 10.000 IU, vitamin D_3_ 900 IU, vitamin E 50.0 mg, vitamin K 2.0 mg, vitamin B_1_ 2.0 mg, folic acid 5.0 mg, pantothenic acid 20.0 mg, vitamin B_6_ 2.0 mg, choline 1200 mg, vitamin B_12_ 0.01 mg, niacin 50 mg, biotin 0.2 mg, Cu 0.1 mg, Fe 75.0 mg, Mn 8.5 mg, Zn 70 mg.

**Table 2 animals-10-02234-t002:** Major chemical compounds of hydrodistilled thyme essential oil (THO) as detected by gas chromatography spectrophotometer (GC-MS).

Chemical Compounds	Rt.	Area %	Mol. Weight (g/mol)	Chemical Formula
p-Cymene	6.99	23.59	134.218	C_10_H_14_
Β-linalool	9.61	0.74	154.25	C_10_H_18_O
Carvone (Carvacrol)	15.70	9.80	150.22	C_10_H_14_O
Anethole	17.49	2.50	148.2	C_10_H_12_O
Thymol	17.70	39.45	150.22	C_10_H_14_O
Carvacrol	18.09	2.07	150.217	C_10_H_14_O
*trans*-Caryophyllene	22.46	0.98	204.36	C_15_H_24_
γ-terpinene	25.14	12.49	136.23	C_10_H_16_
Aromadenrene	34.84	2.12	204.35	C_15_H_24_
Ledol	48.66	2.24	222.358	C_15_H_26_

**Table 3 animals-10-02234-t003:** Effect of nanoparticles-zinc oxide (ZnO-NPs; 100 mg/kg), thyme oil (THO; 500 mg/kg) and their combination (ZnO-NPs; 100 mg/kg + THO 500 mg/kg) on nutrient digestibility of male rabbits. Letters (a, b, c) denote the significant differences between the groups.

Items	Nutrient Digestibility%				*p*-Value
	Treatments (Mean ± SEM)				
	Control	ZnO-NPs	THO	ZnO-NPs + THO	
Feed inatke g/day	155 ± 2.01	157 ± 2.11	158 ± 2.21	159 ± 2.05	0.089
Dry matter	64.70 ± 1.02 ^b^	67.32 ± 1.03 ^a^	68.38 ± 0.71 ^a^	67.04 ± 0.69 ^a^	0.001
Crude protein	78.06 ± 1.13 ^b^	84.30 ± 1.06 ^a^	85.56 ± 0.51 ^a^	84.30 ± 1.33 ^a^	0.001
Ether extract	80.98 ± 0.89 ^b^	87.24 ± 1.55 ^a^	88.40 ± 1.46 ^a^	86.98 ± 0.58 ^a^	0.040
Crude fiber	24.92 ± 1.67 ^b^	27.40 ± 1.02 ^a^	27.91 ± 0.61 ^a^	27.05 ± 1.22 ^a^	0.035
Energy	66.71 ± 1.30 ^b^	67.90 ± 1.34 ^a^	69.52 ± 1.42 ^a^	68.24 ± 1.23 ^a^	0.031

^a–c^ Means not sharing a common superscript in a row are significantly different (*p* < 0.05). Means are average of twenty replicates (*n* = 25 animals per treatment) determined at the end of the experimental period. ZnO-NPs: Nanoparticles of zinc oxide THO: Thyme oil; ZnO-NPs + THO: combination of nanoparticles of zinc oxide and thyme oil; SEM: Standard error of the mean.

**Table 4 animals-10-02234-t004:** Effect of ZnO-NPs, THO and their combination on semen characteristic of male Californian rabbits.

Items	Treatments (Mean ± SEM)				*p*-Value
	Control	ZnO-NPs	THO	ZnO-NPs + THO	
Live sperm %	77.34 ± 0.68 ^b^	83.99 ± 0.84 ^a^	84.20 ± 0.59 ^a^	84.50 ± 0.70 ^a^	0.003
Abnormal sperm %	17.40 ± 0.41 ^a^	15.58 ± 0.51 ^b^	15.10 ± 0.50 ^b^	15.20 ± 0.37 ^b^	0.001
Sperm motility %	56.67 ± 1.51 ^b^	75.00 ± 1.49 ^a^	75.67 ± 0.86 ^a^	73.33 ± 0.68 ^a^	0.002
Semen volume, ml	0.63 ± 0.01 ^c^	0.74 ± 0.01 ^b^	0.72 ± 0.02 ^b^	0.76 ± 0.01 ^a^	<0.05–0.0001

^a–c^ Means not sharing a common superscript in a row are significantly different (*p* < 0.05). Means are average of twenty replicates (*n* = 25 animals per treatment) determined at the end of the experimental period. ZnO-NPs: Nanoparticles of zinc oxide THO: Thyme oil; ZnO-NPs + THO: combination of nanoparticles of zinc oxide and thyme oil; SEM: Standard error of the mean.

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
