# Peer review of "Combined Supplementation of Nano-Zinc Oxide and Thyme Oil Improves the Nutrient Digestibility and Reproductive Fertility in the Male Californian Rabbits"

_animals, 2020, doi:10.3390/ani10122234_

Round 1

Reviewer 1 Report

Dear authors,

In my opinion, the latest version of manuscript can be published in ,,Animals”.

Well done!

Author Response

Dear Reviewer 

Thank you very much for your revision and kind words

Regards

Dr. Ahmed Ezzat

Reviewer 2 Report

The authors attended to all the guests and I think the article has improved substantially in form and content.

I just have some minor corrections that I propose to the authors in order to a better fluidity of the manuscript:

Line 41-43, I think that is better to write, to give an order to ideas: “…. Nutrients digestibility, liver and kidney functions, testosterone concentration, and semen characteristics (sperm motility, vitality, and reduced abnormalities)…”

Line 52 – This error was referred by me in the previous review was not corrected in the abstract: “initial body weight 3.480 ± 75.5”

Line 55 – Remove “respectively”

Line 57, 192, etc– Similarly to the previous suggestion in lines 41-43, authors should try to give an order that makes sense to the readers. In this case “Furthermore, sperm vitality, motility, and semen volume…”. When we evaluate semen the order is : volume, motility and vitality, so I suggest to write in this way.

Line 100 – “was continuously available ad libitum” , remove “continuously”, it is the same that “ad libitum”

Line 114 “Zinc oxide nanoparticles (Cas no.1314-13-2, Sigma-Aldrich, Steinheim, Germany) concentration 114 was 20 wt. % in H2O,”. Using the chemical formula of water is not correct in this case. I suppose you diluted it in normal potable water, not in ultrapure water made up only of H2O molecules. Please replace H2O to “potable water” or the class of water that you used, distillated, miliQ etc

Table 3 : “SEM: Standard error of the means”, replace means to mean

Figure 4 – It is a suggestion, non-obligated. In figure 4 you should replace a photograph with live and dead sperms. The inset of dead sperm is of very bad quality, in disaccording with the quality of your work.

Figure 3 and Fig 4 – You wrote the acronyms for creatinine and testosterone, but you never used it again. Remove the acronyms.

Author Response

Authors´ response: Dear Reviewer, thank you for your helpful comments and evaluation of our manuscript which improved the fluidity in the final form of our paper! Please, find our reply to your comments below and a list of changes that we made according to your suggestions. We have addressed all suggestions which are highlighted by turquoise color.

Q1. Line 41-43, I think that is better to write, to give an order to ideas: “…. Nutrients digestibility, liver and kidney functions, testosterone concentration, and semen characteristics (sperm motility, vitality, and reduced abnormalities)…”

Author response R1. Done as requested

===================================================

Q2. Line 52 – This error was referred by me in the previous review was not corrected in the abstract: “initial body weight 3.480 ± 75.5”

Author response R2. Corrected

===================================================

Q3. Line 55 – Remove “respectively”

Author response R3. Done, removed.

===================================================

Q4. Line 57, 192, etc– Similarly to the previous suggestion in lines 41-43, authors should try to give an order that makes sense to the readers. In this case “Furthermore, sperm vitality, motility, and semen volume…”. When we evaluate semen the order is; volume, motility and vitality, so I suggest to write in this way.

Author response R4. All corrected as: semen volume, sperm motility, vitality, and morphology

===================================================

Q5. Line 100 – “was continuously available ad libitum” , remove “continuously”, it is the same that “ad libitum”

 Author response R5. The sentence is corrected as requested. The word continuously was removed.

===================================================

Q6. Line 114 “Zinc oxide nanoparticles (Cas no.1314-13-2, Sigma-Aldrich, Steinheim, Germany) concentration 114 was 20 wt. % in H2O,”. Using the chemical formula of water is not correct in this case. I suppose you diluted it in normal potable water, not in ultrapure water made up only of H2O molecules. Please replace H2O to “potable water” or the class of water that you used, distillated, miliQ etc

Author response R6. Corrected as requested (Line 114).

===================================================

Q7. Table 3: “SEM: Standard error of the means”, replace means to mean

Author response R7. Corrected as requested (Line 190).

===================================================

Q8. Figure 4 – It is a suggestion, non-obligated. In figure 4 you should replace a photograph with live and dead sperms. The inset of dead sperm is of very bad quality, in disaccording with the quality of your work.

Author response R8. Dear Prof. Reviewer, usually the dead spermatozoa appears in that form demonstrated in figure 4B. We preferred placing the photograph in order to demonstrate all the work especially the spermatozoa which deal with our specialty for promotion. We appreciate your suggestion but please accept it.

===================================================

Q9. Figure 3 and Fig 4 – You wrote the acronyms for creatinine and testosterone, but you never used it again. Remove the acronyms.

Author response R9. Corrected as requested. The acronyms are rempved from the figures of creatinine and testosterone.

Thanks very much for your valuable comments

===================================================

This manuscript is a resubmission of an earlier submission. The following is a list of the peer review reports and author responses from that submission.

Round 1

Reviewer 1 Report

The paper entitled "Combined supplementation of nano-zinc oxide and thyme oil improves the nutrient digestibility and reproductive fertility in the male Californian rabbits" is interesting and falls into the scope of the journal. However, there are some weaknesses in the study related to the absence of some important information. Moreover, the presentation of the results and their discussion should be revised.

General comment

The main objective of the study was the evaluation of the effect of dietary supplementation with nanoparticles-zinc oxide, thyme oil, and their combination on reproductive performance of male rabbits. However, the only variables measured to this aim were the percentages of live sperm, abnormal sperm, sperm motility and the semen volume. The measurement of these variables is a fundamental step for this type of study, but answers to practical questions are lacking: Is the fertility of the male rabbits really improved? And to what extent is it improved?

In this study the dietary treatments decreased serum concentration of ALT and AST. High levels of these enzymes could indicate the presence of cellular damages in the hepatic, cardiac and muscular tissues, however, the absolute levels of ALT and AST reported in this study does not seem so high. Thus, the control animals could be considered as healthy as the treated animals were.

Were the feed intake and the live weight of the animals controlled during the trial? If yes, why the data were not presented and discussed?

Specific comments

L37-38. Reproductive performance of male rabbits cannot be evaluated only in terms of semen quality traits. Moreover, in the study the digestibility of nutrients was not expressed as coefficients. Please revise. The same for L49-50.

L42-45. This sentence is not clear for me, please revise. Furthermore, on the basis of your data, with the exception of testosterone level, there is no synergistic effect between nanoparticles-zinc oxide and thyme oil on the assessed variables.

L67-89. Please motivate how the dietary inclusion levels of nanoparticles-zinc oxide, thyme oil were chosen.

L77. The citation n. 9 is not pertinent. Please delete.

L78-80. This sentence and the related citations are not necessary. Please delete.

L98. Were the animals fed ad libitum?

Table 1. Fibre fractions (aNDF, ADF and ADL) should be reported rather than crude fibre.

Table 1. Was the digestible energy content estimated? According to?

Table 1. The basal content of zinc should be reported.

L137-146. Harmonized procedures for in vivo digestibility trials in rabbits are detailed in Perez et al., 1995.

L142. “Ether extract” not “fat-ether extract”. Please change accordingly in all the manuscript.

L174. You do not show the digestibility data as coefficients.

L176. “but THO was the most effective agent in improving the nutrient digestibility”. I do not see that. Please delete.

L176-178. The direct comparison between control group and THO group is not appropriate. Instead, you should compare the average of treated groups with the control group.

Table 3. Please report the data of feed intake.

Table 3. The digestibility of protein in treated groups seems very high. Please check these data.

Table 3. The digestibility of fibre fractions (aNDF, ADF and ADL) should be reported rather than that of crude fibre.

Table 3. Which was the digestibility of energy?

L194-198. There are no statistically significant differences among treated groups. Therefore, you should compare the average of treated groups with the control group.

Table 4. Please check the superscript letters in the row “Abnormal sperm”. In general the letter “a” was assigned to the higher values. The same should be done for all variables presented.

L204-2012. The presentation of these data should be revised. The authors report in the text only p-values, but these variables deserve further details (e.g. data presentation in absolute terms).

L231-232. Which was the total content of zinc in the diets?

L237. “Simultaneously, in...”

L240. In think that some pertinent citations could be added.

L266-267. I do not agree. In my opinion there are no elements which sustain a synergistic effect between nanoparticles-zinc oxide and thyme oil.

Author Response

Comments and Suggestions for Authors

The paper entitled "Combined supplementation of nano-zinc oxide and thyme oil improves the nutrient digestibility and reproductive fertility in the male Californian rabbits" is interesting and falls into the scope of the journal. However, there are some weaknesses in the study related to the absence of some important information. Moreover, the presentation of the results and their discussion should be revised.

Authors´ response:  Dear Reviewer, thank you for your helpful comments and evaluation of our manuscript which improved the final form of our paper. Please find below is our reply to your comments and a list of changes that we made according to your suggestions. We have addressed all suggestions and highlighted them in yellow colour in the revised manuscript.

General comment

Point 1: The main objective of the study was the evaluation of the effect of dietary supplementation with nanoparticles-zinc oxide, thyme oil, and their combination on reproductive performance of male rabbits. However, the only variables measured to this aim were the percentages of live sperm, abnormal sperm, sperm motility and the semen volume. The measurement of these variables is a fundamental step for this type of study, but answers to practical questions are lacking: Is the fertility of the male rabbits really improved? And to what extent is it improved?

Authors´ response 1We thank the reviewer for this excellent point. We agree that the fertility is a critical point to support the results. Our current results showed improvement of semen characteristics, testosterone levels, and spermatozoal microscopic parameters that are a reliable and preliminary evidence to the improved male fertility in comparison to the control groups. The present results of semen and spermatozoal characteristics represent the core parameters of fertility, while the conception rates and relative factors will be considered in our further study.

========================================================================   

Point 2: In this study, the dietary treatments decreased serum concentration of ALT and AST. High levels of those enzymes could indicate the presence of cellular damages in the hepatic, cardiac and muscular tissues, however, the absolute levels of ALT and AST reported in this study does not seem so high. Thus, the control animals could be considered as healthy as the treated animals were.

Authors´ response 2: There are transient increases in those enzymes away from the diseases as appeared in the control. The physical activities like exercise are enough reasons to increase the concentration levels of those liver enzymes; ALT and AST, in parallel to the healthy state (Giannini  et al. 2005)*. Thus, the decrement of the hepatic enzymes in response to the tested ZnO-NPs and/or THO indicate their effects of the liver enzymes away and keeping them in low levels. Therefore, all animals are considered healthy and the liver enzymes were within the normal ranges.

*Giannini EG, Testa R, Savarino V. Liver enzyme alteration: a guide for clinicians. CMAJ. 2005;172(3):367-379. doi:10.1503/cmaj.1040752

========================================================================

Point 3: Were the feed intake and the live weight of the animals controlled during the trial? If yes, why the data were not presented and discussed?

Author response 3: Thank you very much for your comments, the feed intake data was provided in Table 3, but the growth performance was not the target of the present study as we run the study on adult rabbits, therefore the data was not presented.

Specific comments

Point 4: L37-38. Reproductive performance of male rabbits cannot be evaluated only in terms of semen quality traits. Moreover, in the study the digestibility of nutrients was not expressed as coefficients. Please revise. The same for L49-50.

Author response 4: Dear Reviewer, thank you very much as we mentioned above; improvement of semen characteristics, testosterone levels, and spermatozoal microscopic parameters could be the preliminary evidence to the improved male fertility. The conception rates of male rabbits in response to the present treatments will be considered in our future studies.

According to your comment, we have changed the ‘’reproductive performance’’ in the text of manuscript in to ‘’reproductive parameters’’ and the changes were highlighted yellow. The digestibility of nutrients was expressed as coefficients. Please see Page 5, Line 144-146: Apparent nutrient digestibility coefficients of the diets were determined according to the classical formula (Perez et al., 1995):

========================================================================

Point 5: L42-45. This sentence is not clear for me, please revise. Furthermore, on the basis of your data, with the exception of testosterone level, there is no synergistic effect between nanoparticles-zinc oxide and thyme oil on the assessed variables.

Author response 5: Thank you very much, we revised the sentence and corrected it by deleting the last words of ‘’male fertility’’ and highlighted the corrected sentence yellow.

Line 42-43: Therefore, we recommend inclusion of ZnO-NPs or THO or both for rabbit feeding regimen in improving the feeding profitability.

========================================================================

Point 6: L67-89. Please motivate how the dietary inclusion levels of nanoparticles-zinc oxide, thyme oil were chosen.

Author response 6: The requested information was added. Please see page 3 Line 79-83 (Likewise, supplementation with thyme oil (THO; 0.5 g/kg) was found to improve the intestinal integrity and total antioxidant status of rabbits [17]. The main components of thyme oil are thymol, carvacrol, p-cymene, g-terpinene, linalool, b-myrcene, terpinen-4-ol [13, 18]. Those constituents were known to have antioxidant properties [20] and may improve liver and kidney functions and abdominal fat accumulation [13, 20]).

Please see page 8 Line 240-241. …….The zinc levels in the rabbit rations vary widely according to dietary ingredients and zinc supplementation, ranging from 30 to 170 mg/kg in rabbit diets (36, 37). Since there are almost unlimited possibilities concerning dosage of zinc there is still more research needed to determine the optimum levels and sources of Zn in rabbit diets.

===================================================

Point 7: L77. The citation n. 9 is not pertinent. Please delete.

Author response 7:  Deleted as requested

===================================================

Point 8: L78-80. This sentence and the related citations are not necessary. Please delete.

Author response 8:  Deleted as requested

===================================================

Point 9: L98. Were the animals fed ad libitum?

Author response 9: yes, please see page 3 Line 97-98. The feed and fresh tap water was continuously available ad libitum. The sentence was highlighted yellow.

===================================================

Point 10: Table 1. Fiber fractions (aNDF, ADF and ADL) should be reported rather than crude fiber.

Author response 10:  Sorry the fiber fractions (aNDF, ADF and ADL) were not analysed according to the available sets and machines in our laboratory.

===================================================

Point 11: Table 1. Was the digestible energy content estimated? According to?

Author response 11: The digestible energy (DE) refers to gross energy intake minus energy lost in faeces according to Hall et al. [24] (Line 147).

===================================================

Point 12: Table 1. The basal content of zinc should be reported.

Author response 12:  It is added as requested see Table 1 (Page 3): calculated as Zn 0.09 g/kg, and added as 0.09 %.

===================================================

Point 13: L137-146. Harmonized procedures for in vivo digestibility trials in rabbits are detailed in Perez et al., 1995.

Author response 13: Thanks, we have followed this procedure. See Page 5 Line 143-147: …… At the end of the experiment at 33 weeks of age, a digestibility trial was carried out for four days according to Perez et al. [Error! Reference source not found.]. The DE refers to GE intake minus energy lost in faeces according to Hall et al. [24]

===================================================

Point 14: L142. “Ether extract” not “fat-ether extract”. Please change accordingly in all the manuscript.

Author response 14: Thanks, corrected as requested in all the text of the manuscript.

===================================================

Point 15: L174. You do not show the digestibility data as coefficients.

Author response 15: The digestibility of nutrients was expressed as coefficients. Please see Page 5, Line 143-144: Apparent nutrient digestibility coefficients of the diets were determined according to the classical formula (Perez et al. [23]):

===================================================

Point 16: L176. “but THO was the most effective agent in improving the nutrient digestibility”. I do not see that. Please delete.

Author response 16: Thanks, deleted as requested

===================================================

Point 17: L176-178. The direct comparison between control group and THO group is not appropriate. Instead, you should compare the average of treated groups with the control group.

Author response 17: Thank you for your comment. We have four different treatment groups including control, THO, ZnO-NPs and combination of THO and ZnO-NPs. Therefore, one-way analysis of variance (ANOVA) is the suitable statistical test to determine whether there were any statistically significant differences between the means of control and there different treated groups/variables.

===================================================

Point 18: Table 3. Please report the data of feed intake.

Author response 18: Thank you very much for your comment, the feed intake data was provided in Table 1. Page 3 Line 104, additionally, the feed consumption for each pen between weighing was determined through the measurement of feed residue on the same day as the rabbits were weighed.

Page 6, Line 174 - 188:…. No significant differences were detected between the treated groups in feed intake, but significantly to highly significantly differences were recorded between each nutrient and its respective control values as shown in Table 3.

================================================

Point 19: Table 3. The digestibility of protein in treated groups seems very high. Please check these data.

Author response 18: The digestibility of protein in treated groups was within the normal values (please see Alvarenga et al., Animals 2017, 7, 95; doi:10.3390/ani7120095)

===================================================

Point 20: Table 3. The digestibility of fibre fractions (aNDF, ADF and ADL) should be reported rather than that of crude fiber.

Author response 20: We are sorry for this shortage, the fiber fractions (aNDF, ADF and ADL) were not analyzed.

===================================================

Point 21: Table 3. Which was the digestibility of energy?

Author response 21: Thank you very much for your comments, energy data was provided in Table 3.

===================================================

Point 22: L194-198. There are no statistically significant differences among treated groups. Therefore, you should compare the average of treated groups with the control group.

Author response 22: Thanks; this is what already statistically corrected to determine whether there were any significant differences between the means of control and treated groups. Please look at the table 4, and the statistical analysis as shown in Table 4 (Page 6, 7: Lines: 191-201).

===================================================

Point 23: Table 4. Please check the superscript letters in the row “Abnormal sperm”. In general, the letter “a” was assigned to higher values. The same should be done for all variables presented.

Author response 23: Thank you very much, the letters were revised and statistically corrected in the table (Page 7, Lines: 202 - 204).

===================================================

Point 24: L204-212. The presentation of these data should be revised. The authors report in the text-only p-values, but these variables deserve further details (e.g. data presentation in absolute terms).

Author response 24: According to your valuable comment, all results and figures 2-4, were revised and re-edited again. Please see pages 7, 8 (Lines: 205 – 238).

================================================

Point 25: L231-232. Which was the total content of zinc in the diets?

Author response 25: It is added as requested in page 9, line 250-251: ‘’Zinc requirement for rabbits is 30-170 mg/kg dry matter, with higher requirements for breeders’’

================================================

Point 26: L237. “Simultaneously, in...”

Author response 26: Corrected (Page 9, Line 256)

===================================================

Point 27: L240. In think that some pertinent citations could be added.

Author response 27: Thanks, new citations were added (Lines: 256-261):

[39] Garg, A.K.; Vishal Mudgal, Dass, R.S. Effect of organic zinc supplementation on growth, nutrient utilization and mineral profile in lambs. Anim. Feed Sci. Technol. 2008, 144, 82-96. doi: 10.1016/j.anifeedsci.2007.10.003

[40] Heo, J.M.; Kim, J.C.; Hansen, C.F.; Mullan, B.P.; Hampson, D.J.; Pluske, J.R. Effects of dietary protein level and zinc oxide supplementation on performance responses and gastrointestinal tract characteristics in weaner pigs challenged with an enterotoxigenic strain of Escherichia coli. Anim. Prod. Sci. 2010, 50, 827-836. doi:10.1071/AN10058

[41] Sarvari, B.G.; Seyedi, A.H.; Shahryar, H.A.; Sarikhan, M.; Ghavidel, S.Z. Effects of dietary zinc oxide and a blend of organic acids on broiler live performance, carcass traits, and serum parameters. Braz. J. Poult. Sci. 2015, 17, 39-45. doi: 10.1590/1516-635XSPECIALISSUENutrition-PoultryFeedingAdditives039-046

[42] Mohammadi, V.; Ghazanfari, S.; Mohammadi-sangcheshmeh, A.; Nazaran, M.H. Comparative effects of zinc-nano complexes, zinc-sulphate and zinc-methionine on performance in broiler chickens. Brit. Poult. Sci., 2015, 56, 4, 486-493. DOI: 10.1080/00071668.2015.1064093

[43] Ribeiro, A.D.B.; Ferraz Junior, M.V.C.; Polizel, D.M.; Miszura1, A.A.;  Barroso, J.P.R.;  Cunha, A.R.; Souza, T.T.; Ferreira, E.M.; Susin, I.; Pires, A.V. Effect of thyme essential oil on rumen parameters, nutrient digestibility, and nitrogen balance in wethers fed high concentrate diets. Arq. Bras. Med. Vet. Zootec. 2020, 72, 573-580, doi: 10.1590/1678-4162-11322

===================================================

Point 28: L266-267. I do not agree. In my opinion, there are no elements that sustain a synergistic effect between nanoparticles-zinc oxide and thyme oil.

Author response 28: The conclusion was modified as requested (Page 10)

In view of the above findings, it can be concluded that ZnO-NPs, THO, or their combination in rabbit feeding resulted in improved nutrient digestibility, liver and kidney functions, as well as semen quality. The combined treatment of ZnO-NPs with THO suggested a possible synergistic effect for stimulating testosterone secretion and increasing the volume of seminal plasma. Further molecular analysis and fertility studies are required to elaborate on this combinational effect.

Reviewer 2 Report

Authors presents a study using mineral and essential oils nanoparticles to study the impact in the digestibility and fertility in rabbit male.

The authors are sure if it is correct to use “NZnO”? I’m not sure, and I think that is preferable, “ZnO NPs” or at maximum “nZnO”… N is the chemical symbol for nitrogen and can be confusing.

In the Graphic abstract, for the group of “NZnO” the authors present a flask and a petri dish with a powder. In the group of “NZnO+THO” just the flask is presented. This can suggest to the readers that in the group of “NZnO” a additional powder (petri dish) was used. Please, remove the flask or the powder, I personally prefer the dish with powder.

In the graphic abstract the figure of THO was taken off the internet https://www.indiamart.com/proddetail/thyme-oil-20978432855.html. I’m not sure that the copyright was respected and may cause image rights problems. I have not been looking for other figures (rabbit, petri dish…)  that are relatively "nonspecific" but the authors should take this into attention.

Line 51 “(initial body weight 3.480 ± 75.5)” Please present the measure unit, kg or g?. How can SEM be 75.5? This means that they had rabbits with negative weight and others weighing 78 kg. I know that you would want present as 3480 ± 75.5 g, but please pay attention to this along the manuscript.

Keywords – testosterone should be present in minuscule

Line 95 – Please refer how you randomized the animals. Using the jail number, using the earring number or other?

Line 98 – “respectively” should be removed.

Line 124 – “HV = 80 kV”

Table 2 – MW should be presented in g/mol, not gm/mol

B-linahool – chemical formula is “C10H18O” right?

Carvone – chemical formula is “C10H14O” right?

Anethole, Thymol, Carvacrol and Ledol should present a “O” too, right?

Line “one of the external ear veins”, I suppose that the vein used was the marginal ear vein. The term “external ear veins” do not exist, at by best knowledge. Please replace by marginal ear vein if this was the one used, which is likely, given the large volume.

Line 159 - 3000 rpm, all the centrifugation velocities should be presented in gauge units, in alternative the centrifuge shaft radius has to be displayed

Line 237 “In”

NZnO+THO presents a big difference in the testosterone level but the difference is less notorious in the characteristics of the sperm. Authors should discuss this fact.

Authors should also discuss the cost/benefit of this measures. In fact, what the authors think that is the best option? Suplement with Zn, THO or both taking account the costs? We know that the food is the biggest cost in cuniculture so just small increases in costs represent large expenditures.

A main concern about this study is the number of animals used. How the authors calculated that 100 rabbits would be needed for this study? This may to be clearly presented and calculated taking account previous studies and expected variability of results. Taking account that anesthesia and invasive methods were used, this is absolutely critical.

Some mistakes in the References like “-J, line 23”,  or “-D” line 32.

Author Response

Authors present a study using mineral and essential oil nanoparticles to study the impact on the digestibility and fertility of rabbit males.

Authors´ response: Dear Reviewer, thank you for your helpful comments and evaluation of our manuscript which improved the final form of our paper! Please, find our reply to your comments below and a list of changes that we made according to your suggestions. We have addressed all suggestions.

Q1. The authors are sure if it is correct to use “NZnO”? I’m not sure, and I think that is preferable, “ZnO NPs” or at maximum “nZnO”… N is the chemical symbol for nitrogen and can be confusing.

Author response R1. Done. Thank you for your comment, we changed the ‘’NZnO’’ into “ZnO-NPs’’ throughout the text, tables and figures of the manuscript, and the corrections in the text were highlighted, accordingly.

===================================================

Q2. In the graphic abstract, for the group of “NZnO” the authors present a flask and a petri dish with a powder. In the group of “NZnO+THO” just the flask is presented. This can suggest to the readers that in the group of “NZnO” a additional powder (petri dish) was used. Please, remove the flask or the powder; I personally prefer the dish with powder.

Author response R2. Done. The graphical abstract was corrected as requested.

===================================================

Q3. In the graphic abstract the figure of THO was taken off the internet https://www.indiamart.com/proddetail/thyme-oil-20978432855.html. I’m not sure that the copyright was respected and may cause image rights problems. I have not been looking for other figures (rabbit, petri dish…) that are relatively "nonspecific" but the authors should take this into attention.

Author response R3.  Done. The website was inserted on the graphical abstract as requested.

===================================================

Q4. Line 51 “(initial body weight 3.480 ± 75.5)” Please present the measure unit, kg or g?. How can SEM be 75.5? This means that they had rabbits with negative weight and others weighing 78 kg. I know that you would want present as 3480 ± 75.5 g, but please pay attention to this along the manuscript.

Author response R4. Done. Corrected into 3.48 ± 0.08 kg (Line 91)

===================================================

Q5. Keywords – testosterone should be present in minuscule

 Author response R5. Done. Corrected into testosterone in the keywords

===================================================

Q6. Line 95 – Please refer how you randomized the animals. Using the jail number, using the earring number or other?

Author response R6. The rabbits were housed in individual cages and the cage was the unit for our experiment.

Lines 94-96: Animals were individually reared in a closed building in cages (measuring 48 cm × 55 cm × 38 cm, for width × length × height, respectively) of galvanized wire net, equipped with an automatic drinkers and manual feeders.

===================================================

Q7. Line 98 - “respectively” should be removed.

Author response R7. Done (Line 94).

===================================================

Q8. Line 124 - “HV = 80 kV”

 Author response R8. Done. Corrected and highlighted as requested (Line 123)

===================================================

Q9. Table 2 – MW should be presented in g/mol, not gm/mol

Author response R9. Done. Corrected and highlighted as requested (in Table 2)

===================================================

Q10. B-linahool – chemical formula is “C10H18O” right?

Author response R10.  Done and highlighted (in Table 2)

===================================================

Q11. Carvone – chemical formula is “C10H14O” right?

Author response R11.  Corrected and highlighted (in Table 2)

===================================================

Q12. Anethole, Thymol, Carvacrol and Ledol should present a “O” too, right?

Author response R12.  Done and highlighted (in Table 2)

===================================================

Q13. Line “one of the external ear veins”, I suppose that the vein used was the marginal ear vein. The term “external ear veins” do not exist, at by best knowledge. Please replace by marginal ear vein if this was the one used, which is likely, given the large volume.

Author response R12.  Done. Corrected into ‘’marginal ear vein’’ and highlighted (Lines: 159-160)

===================================================

Q14. Line 159 - 3000 rpm, all the centrifugation velocities should be presented in gauge units, in alternative the centrifuge shaft radius has to be displayed

Author response R14.  Done. Corrected into ‘’1008 g’’ and highlighted (Line: 160)

===================================================

Q15. Line 237 “In” NZnO+THO presents a big difference in the testosterone level but the difference is less notorious in the characteristics of the sperm. Authors should discuss this fact.

Author response R15. The variable effects of the treatment groups on testosterone secretion was re-edited in the section of discussion (Pages 9, 10 - Lines: 279-313) explaining that part of testosterone secretion in response to the tested treatments, and supporting the discussion with new citations. Kindly read it. It was highlighted grey because it is a common comment with another reviewer.

===================================================

Q16. Authors should also discuss the cost/benefit of these measures. In fact, what the authors think that is the best option? Supplement with Zn, THO or both taking account the costs? We know that the food is the biggest cost in cuniculture so just small increases in costs represent large expenditures.

Author response R16. Thank you for your comment, but as you know that the feed ingredients cost can be changed from country to country therefore the economic parameters were not considered.

===================================================

Q17. A main concern about this study is the number of animals used. How the authors calculated that 100 rabbits would be needed for this study? This may to be clearly presented and calculated taking account previous studies and expected variability of results. Taking account that anaesthesia and invasive methods were used, this is absolutely critical.

Author response R17. Thanks for your comment, about the number of animals used, but in this experiment, rabbits were individually reared in cages and each cage considered as one replicate, this mean that 25 replicate for each treatment which is enough for rabbit studies according to published papers in the same journal such as: Naturil-Alfonso et al., 2017 (doi.org/10.4995/wrs.2017.6848), Also čobanová et al., 2018 (World Rabbit Sci. 2018, 26: 241-248), Moreover, Safwat et al., 2015 (Animal Feed Science and Technology 201, 72-79), Volek et al. 2018, (Animal Feed Science and Technology 236, 187-195),  Delgado et al., 2017 (Animal Feed Science and Technology 227, 84-94).

 ==================================================

Q18. Some mistakes in the References like “-J, line 23”, or “-D” line 32.

Author response R18. Thanks for your comment. All references were revised, checked and corrected.

 ==================================================

Reviewer 3 Report

I find the manuscript ,,Combined Supplementation of Nano-Zinc Oxide and Thyme Oil Improves the Nutrient Digestibility and Reproductive Fertility in the Male Californian Rabbits” valuable. Below please read my comments.

  • lines 78-80: this statement is true but does not fit this paragraph. Besides, try to avoid self-citations...
  • line 81: plese delete the world ,,also”.
  • lines 167-170: I don’t understand why authors use such complicated procedure for counting intergroup differences. The correct test for this number of animals is the Tukey test. Besides, authors state here that all values were expressed for 25 animals whereas on line 150 there is talk of 15 animals.
  • Discussion: this section should be complemented by a discussion of the effects of thyme oil on rabbits.

Conclusions: this section should be redone. Note that if the results obtained by the last group do not differ from the results obtained by groups 2 and 3, then it points to no sense to give to rabbits the combination you used (NZnO 100 mg/kg and THO 500 mg/kg). Figure 4, however, needs attention and should be discussed in the previous chapter (authors should try to explain why testosterone level jumped so high in the last group).

Author Response

I find the manuscript, Combined Supplementation of Nano-Zinc Oxide and Thyme Oil Improves the Nutrient Digestibility and Reproductive Fertility in the Male Californian Rabbits” valuable. Below please read my comments.

Authors´ response: Dear Reviewer, thank you for your helpful comments and evaluation of our manuscript which improved the final form of our paper! Please find our reply to your comments below and a list of changes that we made according to your suggestions. We have addressed all suggestions and highlighted green.

Q1. Lines 78-80: This statement is true but does not fit this paragraph. Besides, try to avoid self-citations...

Author response R1: Modified as requested (Lines 77-78)

===================================================

Q2. Line 81: please delete the world, also”.

Author response R2: Done (Line 79).

===================================================

Q3. Lines 167-170: I don’t understand why authors use such complicated procedure for counting intergroup differences. The correct test for this number of animals is the Tukey test. Besides, authors state here that all values were expressed for 25 animals whereas on line 150 there is talk of 15 animals.

Author response R3: Thanks very much for your comment. We again analyzed the data of the four treatment groups including the control group by one-way ANOVA using the recommended Tukey test as post-hoc test for the significance differences among groups. Also, the significant differences were confirmed between each two groups by using student t-test for unpaired/one-tailed data. We thank you very much, the corrected data became matching our findings and smoothly discussed (Lines: 167-171, highlighted green). The number of animals used in all treatments was 25, it was corrected and highlighted green (Line 151). Thanks.

===================================================

Q4. Discussion: This section should be complemented by a discussion of the effects of thyme oil on rabbits.

Author response R4: Thank you very much. According to your comment, we added a new paragraph on the effect of feeding thyme oil in rabbits with 2 new reference citations (28, 29) at the beginning of the discussion part (Lines: 240-247) and highlighted with green.

===================================================

Q5. Conclusions: This section should be redone. Note that if the results obtained by the last group do not differ from the results obtained by groups 2 and 3, then it points to no sense to give to rabbits the combination you used (NZnO 100 mg/kg and THO 500 mg/kg). Figure 4, however, needs attention and should be discussed in the previous chapter (authors should try to explain why testosterone level jumped so high in the last group).

Author response R5: Thanks. The part of conclusion was corrected and redone according to the results. We have re-edited as following:

In view of the above findings, it can be concluded that ZnO-NPs, THO, or their combination in rabbit feeding resulted in improved nutrient digestibility, liver and kidney functions, as well as semen quality. The combined treatment of ZnO-NPs with THO suggested a possible synergistic effect for stimulating testosterone secretion and increasing the volume of seminal plasma. Further molecular analysis and fertility studies are required to elaborate on this combinational effect.

Also, the variable effects of the treatment groups on testosterone secretion was reported in the section of results (Lines: 286-313) and highlighted grey. Kindly read it.
